# Inhibition of Na$^+$/K$^+$ ATPase blocks Zika virus infection in mice

Jiao Guo[1,2,4], Xiaoying Jia[1,2,4], Yang Liu[1], Shaobo Wang[1,2,3], Junyuan Cao[1,2], Bo Zhang[1,2], Gengfu Xiao[1,2] & Wei Wang 📧 [1,2✉]

Zika virus (ZIKV) is an infectious disease that has become an important concern worldwide, it associates with neurological disorders and congenital malformations in adults, also leading to fetal intrauterine growth restriction and microcephaly during pregnancy. However, there are currently no approved vaccines or specific antiviral drugs for preventing or treating ZIKV infection. Here, we show that two FDA-approved Na$^+$/K$^+$-ATPase inhibitors, ouabain and digoxin, can block ZIKV infection at the replication stage by targeting Na$^+$/K$^+$-ATPase. Furthermore, ouabain reduced the viral burden of ZIKV in adult mice, penetrated the placental barrier to enter fetal tissues, and protected fetal mice from ZIKV infection-induced microcephaly in a pregnant mouse model. Thus, ouabain has therapeutic potential for ZIKV.

[1] State Key Laboratory of Virology, Wuhan Institute of Virology, Center for Biosafety Mega-Science, Chinese Academy of Sciences, 430071 Wuhan, China. [2] University of the Chinese Academy of Sciences, 100049 Beijing, China. [3] Present address: Shaobo Wang, Department of Pediatrics, University of California San Diego, La Jolla, CA 92093, USA. [4] These authors contributed equally: Jiao Guo, Xiaoying Jia. ✉email: wangwei@wh.iov.cn

Flaviviruses, belonging to the genus *Flavivirus* and family *Flaviviridae*, comprise over 70 different pathogens and are transmitted mostly by arthropods. The emerging and re-emerging flaviviruses, such as Zika virus (ZIKV), Japanese encephalitis virus, dengue virus, West Nile virus, and yellow fever virus, cause public health problems worldwide[1]. Like other closely related flaviviruses, ZIKV contains a single-stranded, positive-sense RNA genome of approximately 11 kb in length, which encodes a polyprotein that is cleaved by host and viral proteases into three structural proteins (capsid, pre-membrane, and envelope) and seven non-structural proteins (NS1, NS2A, NS2B, NS3, NS4A, NS4B, and NS5)[2]. The structural proteins play roles in virus particle formation, receptor binding, virus fusion, and host cell entry; the non-structure proteins are responsible for viral genome replication and host immunity evasion[3,4]. Unlike other flaviviruses, ZIKV can spread directly from person to person through sexual transmission and vertically from mother to fetus[5]. ZIKV infection is related to placental insufficiency, fetal death, microcephaly, and other congenital malformations in fetuses and newborn infants[6], as well as Guillain–Barré syndrome in adults[7].

To date, there is no vaccine or specific approved antiviral drug against ZIKV; thus, there is an urgent need to explore novel antiviral targets and develop effective compounds. Ouabain and digoxin are FDA-approved inhibitors that target $Na^+/K^+$-ATPase. Here, we investigated their antiviral effect on ZIKV both in vitro and in vivo.

## Results

**Antiviral effects of ouabain and digoxin**. To verify the antiviral effects of ouabain and digoxin, we investigated their effects on different ZIKV strains using various cell types. Ouabain and digoxin robustly inhibited virus production, with a reduction of approximately 4–5 log units at the highest concentration (Fig. 1a, c, e–h). A sharp decrease in ZIKV RNA levels was also detected (Fig. 1b, d), indicating strong inhibition of viral replication. Consistent with this inhibition, expression of the viral structural protein NS3 was hardly detectable following treatment with the indicated concentration of digoxin or ouabain (Fig. 1i, j). Moreover, the 50% inhibitory concentration ($IC_{50}$) of both compounds at nanomolar level (Fig. 1k). Together, our results confirm that ouabain and digoxin inhibited ZIKV infection in a dose-dependent manner in vitro.

**Ouabain and digoxin block viral RNA synthesis**. Next, we performed a time-of-addition experiment with ZIKV to determine the stage at which the viral cycle is blocked. Neither ouabain nor digoxin inhibited ZIKV in virucidal or pre-treatments (Fig. 2a). However, both drugs inhibited ZIKV infection during treatment. Most importantly, both ouabain and digoxin showed strong inhibitory effects on ZIKV infection when added post-treatment, suggesting that viral replication was the main stage at which two drugs exerted their inhibitory activity.

To confirm this hypothesis, we investigated the inhibitory effects of ouabain and digoxin on ZIKV replicons. Our results showed that both drugs inhibited ZIKV RNA synthesis in a dose-dependent manner, whereas neither inhibited the initial translation of replicon RNA (Fig. 2b), confirming that these drugs inhibit ZIKV infection at the replication stage.

**ZIKV inhibition with ouabain and digoxin via $Na^+/K^+$-ATPase**. To determine whether blockade of $Na^+/K^+$-ATPase is responsible for ZIKV inhibition, infected cells were treated with two compounds and dimethyl sulfoxide (DMSO) and the indicated concentration of NaCl and KCl. Treatment of cells with only digoxin or ouabain led to a dose-dependent decrease in

ZIKV infection. Furthermore, when treated together with increasing extracellular $Na^+$, the inhibitory effects of digoxin or ouabain increased in a dose-dependent manner (Fig. 3a, b). Addition of extracellular $K^+$ with either drug alleviated the inhibition effect (Fig. 3c, d). These data show that the antiviral effects of ouabain and digoxin are positively correlated with extracellular NaCl but inversely correlated with extracellular KCl. Thus, ouabain and digoxin inhibit ZIKV infection by targeting $Na^+/K^+$-ATPase.

**Ouabain decreases ZIKV viral load in adult mouse brain**. To evaluate whether ouabain protects against ZIKV infection in vivo, 6- to 7-week-old type I IFN signaling knockout line ($Ifnar1^{-/-}$) mice were treated with $10^4$ plaque-forming units (PFU) of ZIKV strain H/PF/2013 with 2 mg/kg ouabain. Ouabain reduced the viral load in the brain, whereas it showed mild effects on viral loads in the serum (Fig. 4a). Histological brain alterations revealed perivascular cuffing and glial nodules in ZIKV-infected mice, whereas ouabain treatment alleviated these phenomena (Fig. 4b). Positive cells (ZIKV and GFAP colocalized signals) were extensively detected in the hippocampus, cerebellum, and pons/medulla in the vehicle-treated group, whereas the ZIKV signal was absent in the ouabain-treated group (Fig. 4c).

**Effect of ouabain treatment in maternal and fetal tissues**. To determine whether ouabain can protect against vertical transmission of ZIKV, pregnant C57BL/6 mice were administered anti-*Ifnar* antibody at −1, +1, and +3 days relative to ZIKV infection, and then treated with ouabain at 3 mg/kg for 5 days (Fig. 5a). At day 5 post-infection, fetuses in the vehicle group exhibited both growth restriction and resorption phenotypes (Fig. 5b, black arrow), whereas no such reduction was observed in ouabain-treated animals. A ~21% rate of fetal demise was observed in the vehicle-treated group, whereas ouabain treatment substantially improved fetal outcomes in terms of survival rate (~96%) and fetus size (Fig. 5b–e). Consistent with the phenotypes, after ouabain treatment, the viral burdens in the fetal brain and placenta were reduced by approximately 20-fold ($P < 0.001$) and 2.6-fold ($P < 0.05$), respectively (Fig. 5f, g).

To confirm the protective effects of ouabain treatment, placental tissues and fetal brains were examined by histological analysis. In addition to the decrease in viral load, placental damage was also alleviated by ouabain. ZIKV infection caused severe injury of the placenta, leading to the loss of trophoblasts and reductions in the labyrinth layer size; these damages were not observed in the ouabain-treated group (Fig. 5h, i). Similarly, histopathological assessment of the fetal brains demonstrated that cortical thickness was decreased in the vehicle-treated mice, whereas it remained normal in ouabain-treated mice. As ZIKV shows tropism for proliferative neural progenitor cells (NPCs) and downregulates the expression of NPC markers such as SOX2, we measured SOX2 levels after ouabain treatment. Consistently, the decrease in SOX2-positive cells after ZIKV infection was relieved by ouabain treatment, demonstrating that developmental defects were substantially reduced by ouabain treatment of ZIKV-infected pregnant mice (Fig. 5j–l). These findings indicate that ouabain can substantially reduce ZIKV-associated injuries in the fetal brain and thus prevent the development of microcephaly.

## Discussion

ZIKV is a concerned global health threat because of its wide transmission and is associated with severe neurological and congenital disorders. This virus has rapidly spread to more than 70 territories and countries. Although anti-ZIKV therapies are

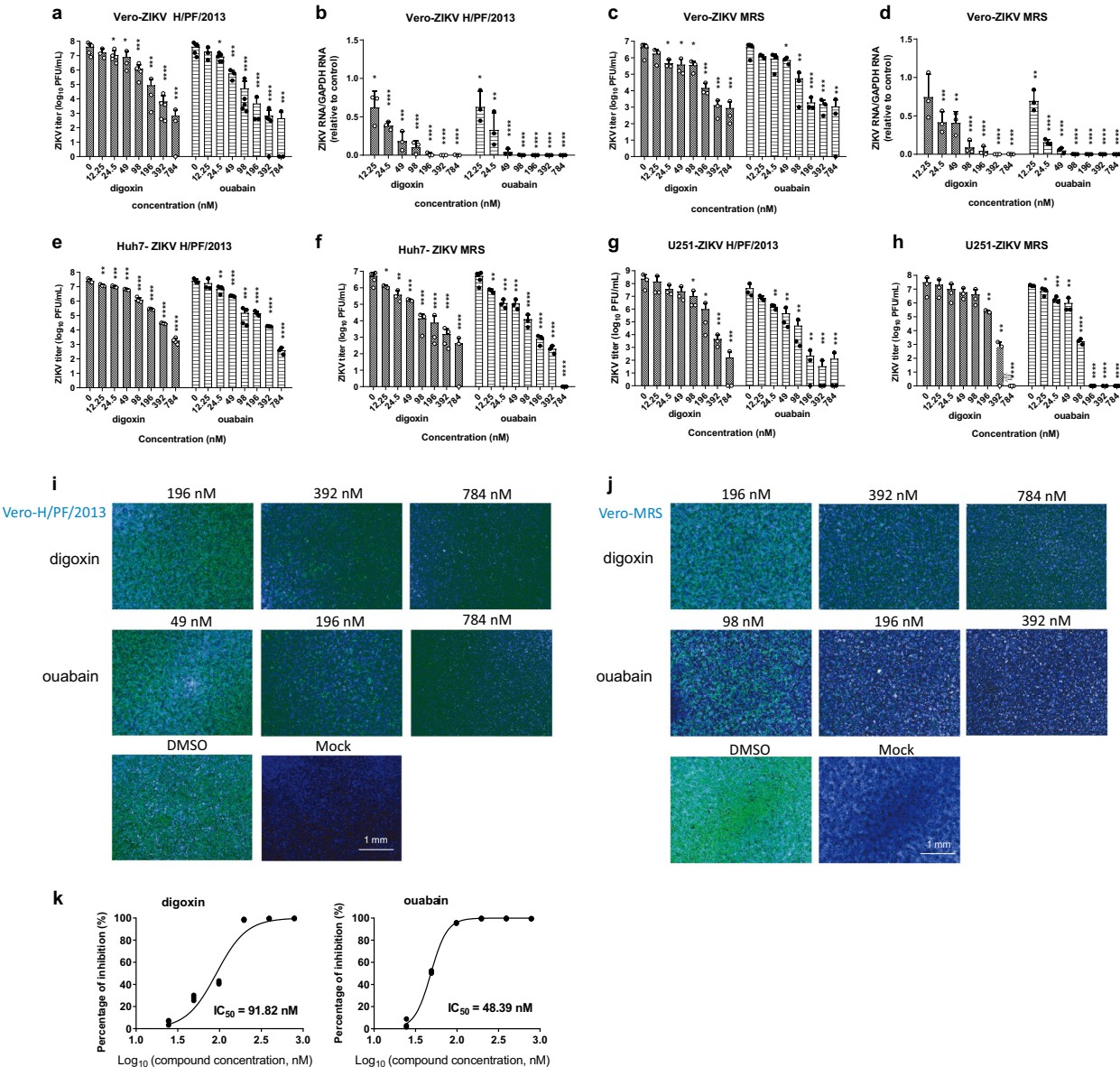

**Fig. 1 Validation of the antiviral effects of ouabain and digoxin. a–h** Antiviral effects of digoxin and ouabain against two ZIKV strains in different cell lines. The cell culture supernatants were subjected to a viral titer assay (**a**, **c**, **e–h**), whereas the cell lysates were assessed by qRT-PCR (**b**, **d**), and duplicate cells were analyzed for protein expression by immunofluorescence staining assay (IFA) (**i**, **j**). IFA images showing the viral protein (green) and nuclei (blue) in Vero cells. **k** Dose–response curves of ouabain and digoxin for inhibition of ZIKV infection in Vero cells. Data are represented as the means ± SDs from at least three independent experiments. ****$P < 0.0001$, ***$P < 0.001$, **$P < 0.01$, *$P < 0.05$.

urgently needed, there are currently no specific vaccines or antiviral drugs for the prevention or treatment of ZIKV infection. Our study suggests that ouabain and digoxin inhibit ZIKV infection.

Ouabain and digoxin are cardiac glycosides used to increase the contraction force of the heart muscle in patients with congestive heart failure and stabilize heart rhythm in patients with atrial arrhythmias. Recently, two drugs were reported to have spectrum antiviral activity by inhibiting chikungunya virus, coronaviruses, human cytomegalovirus, lymphocytic choriomeningitis virus, vesicular stomatitis virus, and Japanese encephalitis virus infection[8–13]. We investigated the antiviral activities of these drugs on ZIKV and found that both ouabain and digoxin robustly inhibited ZIKV infection at a nanomolar level in vitro. Additionally, the ZIKV inhibition effects of ouabain and digoxin are not attributable to cytotoxicity, indicating a large

difference between their therapeutic and toxic concentrations[14]. Moreover, ZIKV can replicate in the central nervous system, and infections result in breakdown of the blood–brain barrier (BBB) along with an influx of inflammatory cells. ZIKV infection was shown to be largely confined to astrocytes in an adult $Ifnar1^{-/-}$ mouse model[15,16]. Glial cells express the ATPase α1 and α2 isoforms, whereas neurons express the α1 and α3 isoforms[17]. In our study, ouabain exhibited therapeutic effects on ZIKV infection in an adult mouse model by decreasing viral loads and alleviating pathological injuries in the brain. As the murine ATPase α1 isoform is less sensitive than its human counterpart, the murine α2 and α3 isoforms were considered as targets of ouabain in the central nervous system[18–20]. Breakdown of the BBB caused by ZIKV infection may have allowed BBB-non-permissive ouabain to enter the brain, bind ATPase, and inhibit viral replication in neurons or glial cells.

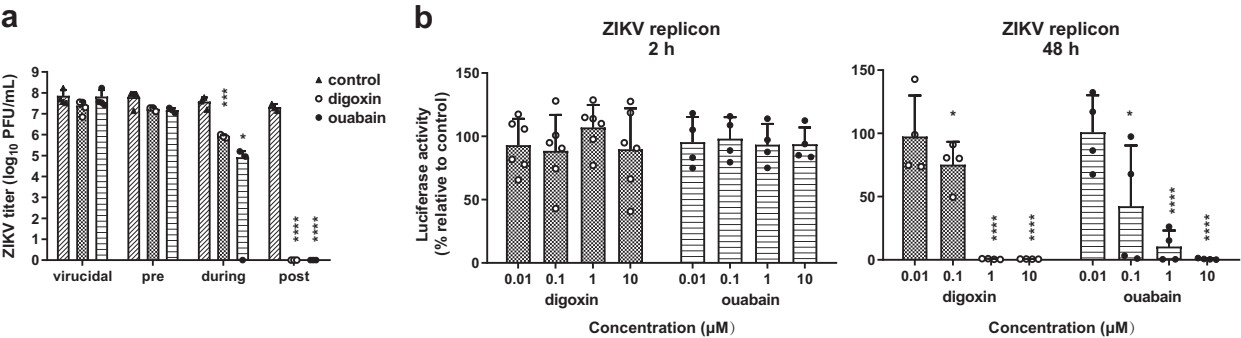

**Fig. 2 Time-of-addition analysis of antiviral activities of ouabain and digoxin. a** Antiviral effects of digoxin and ouabain in each group, determined by the plaque assay. **b** Huh-7 cells were electroporated with the ZIKV replicon were treated with digoxin and ouabain at the indicated concentration, respectively, and luciferase activities were determined as indicated. Data are represented as the means ± SDs from at least three independent experiments. ****$P <$ 0.0001, ***$P <$ 0.001, *$P <$ 0.05.

**Fig. 3 Inhibition of ZIKV by digoxin and ouabain occur via Na⁺/K⁺-ATPase.** Vero cells infected with ZIKV (H/PF/2013) were treated with increasing concentrations of NaCl (**a**, **b**) and KCl (**c**, **d**) at 1 h pre-infection. Cell supernatants were collected for the plaque assay at 48 h post-infection. Data are represented as the means ± SDs from at least three independent experiments. ****$P <$ 0.0001, ***$P <$ 0.001, **$P <$ 0.01, *$P <$ 0.05.

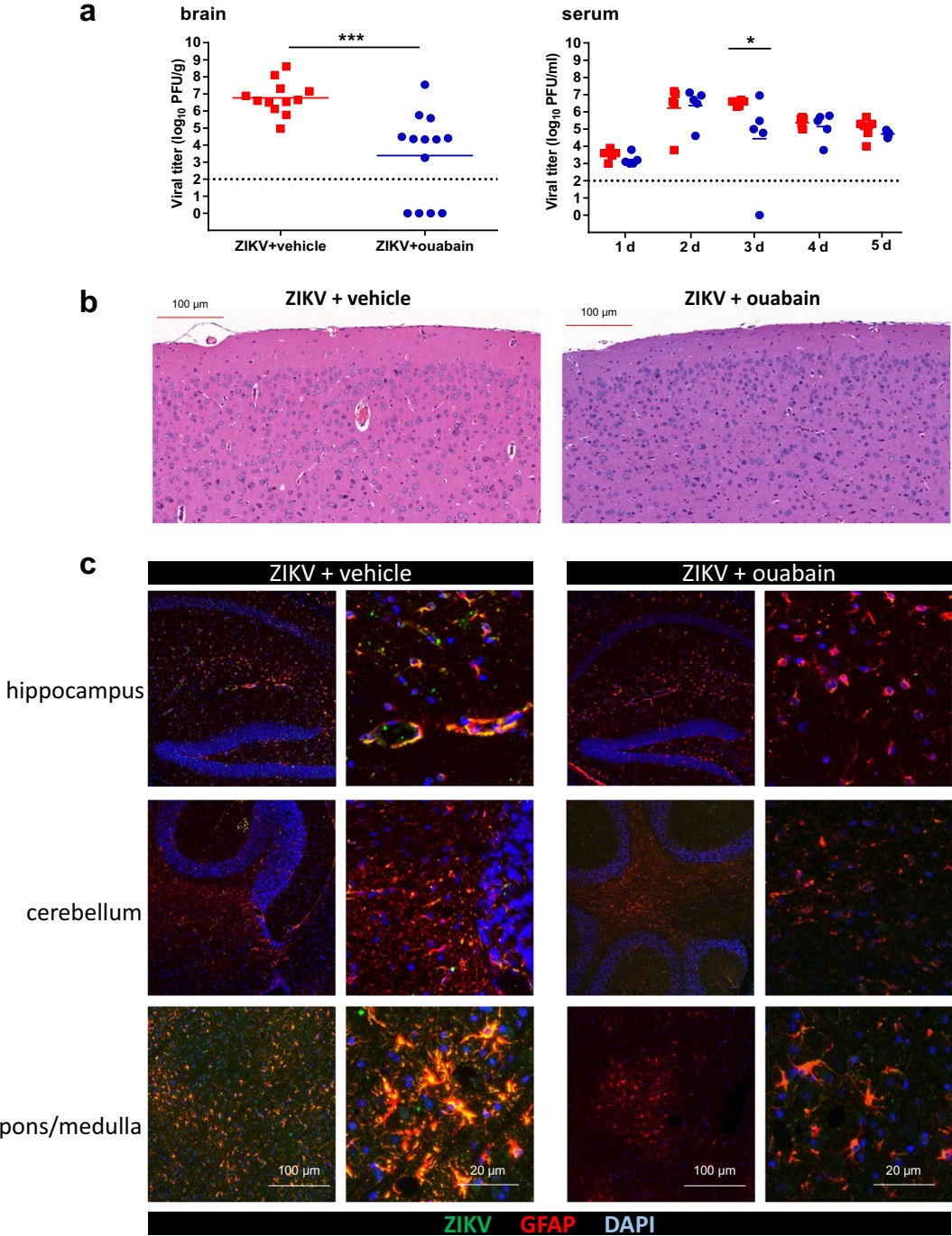

**Fig. 4 Ouabain decreased the ZIKV viral load in adult mouse brains.** *Ifnar1$^{-/-}$* mice were intraperitoneally infected with $1 \times 10^4$ PFU of ZIKV together with ouabain or vehicle. **a** Viral loads in mouse brains and serum were measured by plaque assay on the indicated days. Dashed lines indicate the limit of detection. ***$P < 0.001$, **$P < 0.01$, *$P < 0.05$. **b** Ouabain treatment alleviated the ZIKV infection-induced histopathological injuries in mice. **c** ZIKV infection of astrocytes in mouse brain. Sections of the brain, specifically the hippocampus, cerebellum, and pons/medulla, were stained with antibodies against astrocytes (GFAP), and ZIKV. Nuclei are depicted by DAPI stain (blue).

ZIKV infection during pregnancy is characterized by disruption of placental cells and dysregulation and dysfunction of fetal NPCs, which results in intrauterine growth restriction[21,22]. In this study, we demonstrated that ouabain successfully reduced the viral burden, inhibited the loss of trophoblasts and NPCs, and prevented pathological injuries in mouse placentas and fetal brains, suggesting that ouabain can be used to treat ZIKV infection in pregnant woman and for preventing congenital brain developmental abnormalities caused by perinatal ZIKV infection.

Together, these findings provide valuable information for future clinical trials examining the effects of ouabain in ZIKV infection. Moreover, our results indicate that ATPase is a promising pharmacological target in ZIKV infection.

## Methods
**Ethics statements and mice**. All animal experimental procedures were carried out according to ethical guidelines and were approved by the Animal Care Committee of Wuhan Institute of Virology (Permit Number: WIVA25201801).

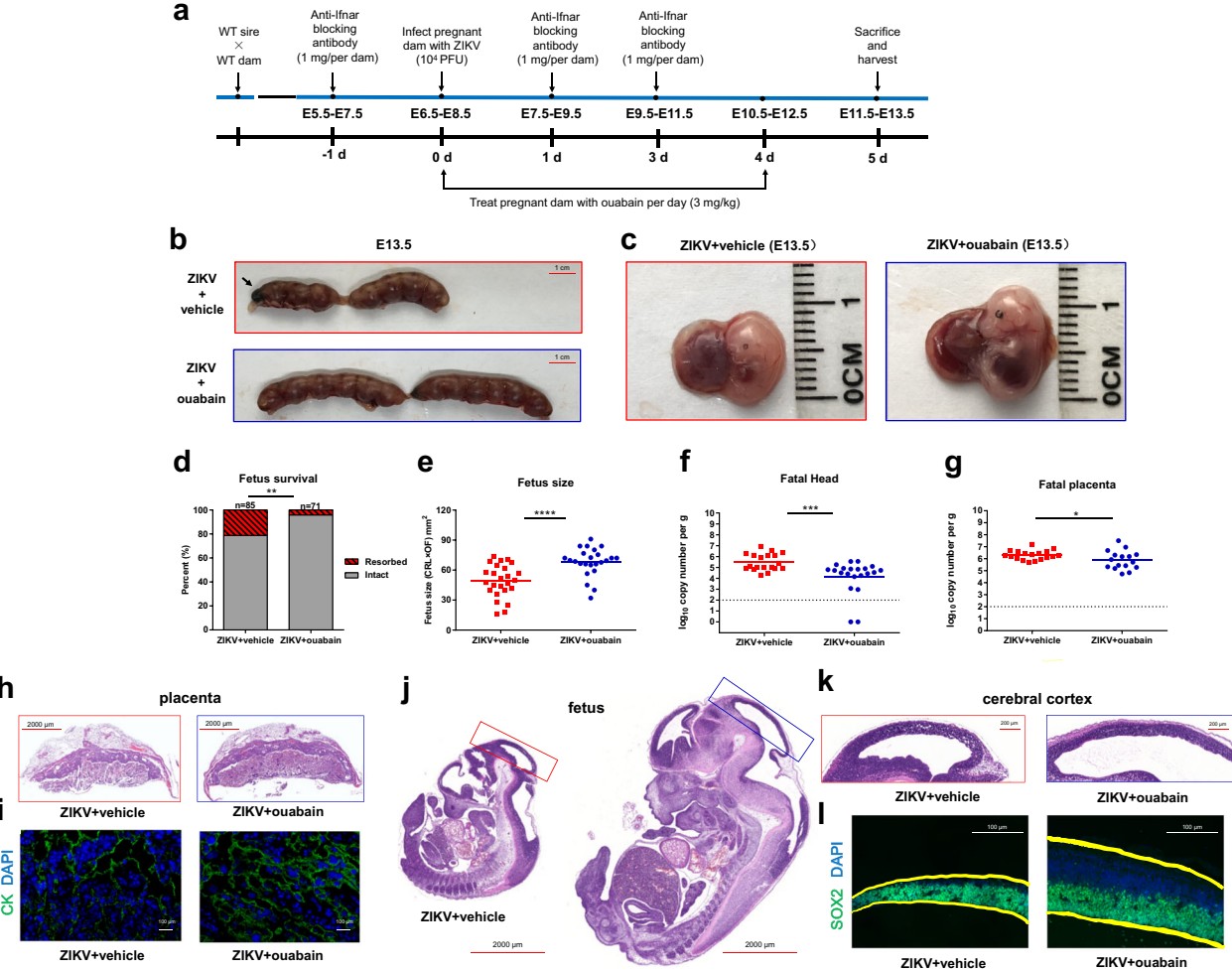

**Fig. 5 Ouabain protected pregnant C57BL/6 mice and their developing fetuses. a** Schematic depiction of ouabain treatment. **b** E13.5 uteri from ouabain-treated and vehicle-treated anti-*Ifnar* dams. **c** Representative images of fetuses from ouabain and vehicle-treated group. **d** Fetus survival on E11.5–13.5 after infection with ZIKV at E6.5–8.5. Data are representative of at least three independent experiments with one pregnant female dam per experiment. The *n* for each group is indicated above each bar. **\*\*P < 0.01. e** Fetus size, as assessed by crown–rump length × occipital–frontal diameter at day 5 post-infection. Bars indicate the mean size of 7–10 fetuses from three independent experiments from fetuses carried by 2–3 pregnant dams. **\*\*\*\*P < 0.0001. f, g** qRT-PCR analysis of viral burden in fetal head and placenta on E11.5–13.5 after infection at E6.5–8.5. Symbols represent individual fetuses collected from several independent experiments. Dashed lines, limit of detection. **\*P < 0.05; \*\*\*P < 0.001. h** Ouabain alleviated reductions in the size of the labyrinth layer of placentas caused by ZIKV infection. **i** Ouabain blocked the loss of trophoblasts caused by ZIKV infection. Immunofluorescence staining of cytokeratin (CK) in mouse placentas. **j–l** Ouabain protected embryonic brains from ZIKV infection and microcephaly. Low magnification images of hematoxylin and eosin (H/E) staining of vehicle- and ouabain-treated fetuses on E13.5 (**j**). High magnification images of H/E staining of cerebral cortex of vehicle- and ouabain-treated fetuses on E13.5 (**k**). Images of cortices stained for NPCs (green, SOX2) and DAPI (blue) (**l**).

**Cells and virus**. Vero, Huh-7, and U251 cells were maintained in Dulbecco's modified Eagle's medium and minimum essential medium containing 10% fetal bovine serum, respectively. The ZIKV strains H/PF/2013 (GenBank accession no. KJ776791.2) and MRS_OPY_Martinique_PaRi_2015 (denoted as MRS, GenBank accession no. KU647676.1) were kindly provided by European Virus Archive Goes Global. The genome sequence of ZIKV strain SZ-WIV001 (GenBank accession no. KU963796) was used as the template for the construction of ZIKV replicon.

**Antiviral compounds**. Ouabain and digoxin were purchased from Sigma-Aldrich (St. Louis, MO, USA).

**Antiviral effects of ouabain and digoxin in vitro**. Vero, Huh-7, and U251 cells in 96-well plates were infected with ZIKV strain H/PF/2013 or MRS at a multiplicity of infection of 0.8 in the presence of different concentrations for 48 h. The antiviral effects of ouabain and digoxin were evaluated by plaque assay, quantitative reverse transcription-PCR (qRT-PCR), and immunofluorescence staining assay (IFA), as previously reported[23,24]. The antibodies used for IFA were as follows: primary antibody anti-ZIKV NS3 (gifted from Dr. Andres Merits, University of Tartu, Estonia) and DyLight[TM] 488 labeled goat anti-rabbit IgG (KPL, Gaithersburg, MD, USA). Nuclei were counterstained with DAPI (Sigma-Aldrich).

**Ouabain and digoxin inhibition of Na⁺/K⁺-ATPase**. Vero cells in 96-well plates were incubated with DMSO or either drug in the presence of increasing concentrations of NaCl and KCl at 1 h pre-infection. The cells were then infected with ZIKV (H/PF/2013) at a multiplicity of infection of 0.8 for 1 h. At 48 h post-infection, the cell supernatants were collected for a plaque assay. The inhibition rate was calculated as the percentage of infected cells normalized to DMSO-treated cells in triplicate experiments.

**Time-of-addition experiment**. To determine which stage of the ZIKV cycle was blocked by each drug, a time-of-addition experiment was performed as previously described[23]. Vero cells were infected with ZIKV strain H/PF/2013 at a multiplicity of infection of 0.8 for 1 h (0–1 h). Ouabain (10 μM) and digoxin (10 μM) were incubated with the cells at the following time points: pre-infection (−1 to 0 h), during infection (0–1 h), and for 47 h post-infection (1–48 h). To exclude a possible direct inactivating effect of the two drugs, viruses were incubated with each drug (1 μM) at 37 °C for 1 h, and the mixtures were diluted by 100-fold to infect Vero cells. Forty-eight hours later, virus titers were determined by performing a plaque assay.

To ensure the effectiveness of ouabain and digoxin in inhibiting ZIKV replication, Huh-7 cells were electroporated with the ZIKV replicon and then incubated with the indicated concentration of either drug, after which Renilla

luciferase activity in the cell lysates was measured using the Rluc system (Promega, Madison, WI, USA).

**Ouabain administration to ZIKV-infected adult mice**. Six- to seven-week-old adult male $Ifnar1^{-/-}$ mice were randomly assigned into two groups and intra-peritoneally infected with $1 \times 10^4$ PFU ZIKV (H/PF/2013). Three hours later, the infected mice were intraperitoneally administered 2 mg/kg ouabain ($n = 13$) or vehicle control ($n = 12$). Treatments were consecutively administered once per day for up to 5 days. The viral burden in the brain and serum was measured by performing a plaque assay. Brain sections were further stained with anti-ZIKV NS3 rabbit serum, glial fibrillary acidic protein (GFAP, Dako, Glostrup, Denmark), and DAPI.

**Ouabain administration to ZIKV-infected pregnant mice**. C57BL/6 background mice were housed under a 12-h light/12-h dark cycle in the Laboratory Animal Center of Wuhan Institute of Virology, Chinese Academy of Sciences (Wuhan, China). For mating, 6- to 7-week-old male and nulliparous female mice were co-housed from 18:00 h to 08:00 h. Day 0.5 of pregnancy was defined as the first observation of a vaginal plug. Pregnant dams were inoculated with ZIKV (H/PF/2013) via the intraperitoneal route with $1 \times 10^4$ PFU of ZIKV in a volume of 100 μL at embryonic days E6.5–E8.5. Infected dams were treated with 1 mg of anti-*Ifnar* antibody (MAR1-5A3, purchased from Leinco Technologies, Inc., St. Louis, MO, USA) on days −1, +1, and +3 relative to ZIKV infection. The infected mice were intragastrically administered ouabain (3 mg/kg of body weight) or vehicle control every day for 5 days post-infection. At day 5, the placentas and fetuses were harvested from the infected mice and ZIKV RNA levels were determined by qRT-PCR. Fetus size was measured as the crown–rump length and occipital–frontal diameter.

**Hematoxylin–eosin staining and immunofluorescence**. Placentas and fetuses were removed by Caesarian section on E13.5, post-fixed in 4% paraformaldehyde at room temperature, and embedded in paraffin. Placentas and fetuses from different litters with the indicated treatments were sectioned and stained with hematoxylin and eosin to assess morphology. For immunofluorescence staining of the mouse placentas and fetuses, the primary antibodies cytokeratin (1:100, ab52625, Abcam, Cambridge, UK) and Sox2 (1:100, Mouse, ab92494, Abcam) were used. Secondary antibody (1:400, Affinity Biosciences, Cincinnati, OH, USA) was applied for 1 h at 37 °C. The sections were then washed with phosphate-buffered saline and counterstained with DAPI.

**Statistics and reproducibility**. In vivo and vitro experiments included contain at least three independent replicates, respectively. These experiments were performed by experiment experience, preliminary experiments, and referenced literature. Student's *t*-test was used to evaluate the statistical significance of differences. A value of $P < 0.05$ was considered statistically significant.

**Reporting summary**. Further information on research design is available in the Nature Research Reporting Summary linked to this article.

## Data availability

All data supporting the findings of this study are available within the paper and its supplementary files or available from the corresponding author upon reasonable request. Source data underlying plots shown in figures are available in Supplementary Data 1.

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

## Acknowledgements

We thank the Center for Instrumental Analysis and Metrology, Core Facility and Technical Support, and Center for Animal Experiment, Wuhan Institute of Virology, for providing technical assistance. This work was supported by the National Key Research and Development Program of China (2018YFA0507204) and National Natural Sciences Foundation of China (31670165).

## Author contributions

J.G., X.J., and W.W. conceived and designed the study. Experiments were performed by J.G., X.J., Y.L., S.W., and J.C. Data were analyzed by J.G. and X.J. Part of the materials were provided by B.Z. Advice for the project was received from G.X. W.W. and J.G. wrote the manuscript. All authors read and approved the final manuscript.

## Competing interests

The authors declare no competing interests.
