## [Peer Review File · Communications Biology]

Reviewers' comments:

Reviewer #1 (Remarks to the Author):

The manuscript by Guo et al study the role that inhibitors of the Na,K-ATPase have in Zika virus replication. The manuscript is extremely short with 3 out of 5 figures as supplemental data as well as part of the discussion as supplemental.

Although there is a fairly amount of studies showing that cardiac glycosides have anti-viral effects and there is a manuscript studying the role of cardiac glycosides in Dengue Virus, another Flavivirus (Cheung, Antiviral Res, 2014), the present manuscript shows intriguing in vivo data. However, the correlation between the in vitro and the in vivo data in regards to the action of cardiac glycosides is not clear, mechanistic data is lacking and for the results presented, both effects could be by different mechanisms.

Comments:

- The present manuscript does not recognize in the introduction or in the discussion the fact that cardiac glycosides have been already determined to have anti-viral effects in different virus families.
- The mechanism of action of the drugs is important to know. The introduction lacks some explanation about the virus cycle, so the reader can understand in what step the drug might act.
- Cardiac glycosides are toxic (being the Na,K-ATPase a housekeeping protein); the authors should show time of incubation with ouabain and digoxin in supplemental figure 1 as well as cell death at that time.
- It is known that these drugs inhibit protein translation at certain dose. Is this the mechanism that the authors are suggesting? Please, explain and demonstrate if that is the case or another mechanism is proposed.
- Supplemental Figure 3 is very confusing. The cells were pre-treated with different amounts of Na⁺ or K⁺ previous infection. What is the rational? Zero extracellular K⁺ should have the same result as Na,K-ATPase inhibition, as it is the substrate for the pump to work. If I understand the graph correctly at zero extracellular K⁺, there is no replication inhibition.
- What is the effect of ouabain in the mice? Do they get sick? Did the authors measure weight loss?
- The authors claim that the effect observed in the brain is due to Na,K-ATPase alpha2 and alpha 3 inhibition. However, all the effects studied in vitro were in cells that express almost exclusively the alpha1 subunit. Could the mechanism be different with the results in vitro and in vivo?
- What Na,K-ATPase isoforms does the placenta express?

Reviewer #2 (Remarks to the Author):

In this study, the authors found that inhibition of Na⁺/K⁺ ATPase by using ouabain (or digoxin) can block Zika virus (ZIKV) infection in both cell culture and mouse models. Although several studies have reported that those two inhibitors can impairs infections with different viruses such as Chikungunya, Japanese encephalitis virus and human cytomegalovirus with similar mechanisms,

this manuscript is still may afford a promising therapeutic potential in combating ZIKV. The manuscript needs a minor revision before accepted.

1. Please calculate the EC50 value of each inhibitor and clarify the fact that ZIKV inhibition by ouabain (or digoxin) are not attributable to their cytotoxicity.
2. It was reported that cardiac glycosides like ouabain and digoxin could inhibit Na⁺/K⁺ ATPase by binding to the catalytic subunit but did so with less efficiency to specific murine isoforms relative to their human counterparts. Considering the author used the mouse model in this study, please use some murine cell lines to clarify the possible species-specific inhibition effect of ouabain and digoxin providing useful information for justifying the dose usage (2 mg/kg) of ouabain in mouse model.
3. The author proposed ouabain inhibit ZIKV infection at the replication stage, but they only checked the expression of the viral structural protein NS3. Please determine the fate of some other protein machineries involved in different steps of replication stage like NS1 before and after administration of inhibitors or even generating or using some ouabain (or digoxin)-resistant viral mutants to give more clues about the detailed mechanism of those inhibitors.

Response to reviewers.

Review 1:

We do appreciate your professional comments about our work. We have revised and improved our manuscript according to your comments. Our responses to the specific comments are as follows.

The manuscript by Guo et al study the role that inhibitors of the Na,K-ATPase have in Zika virus replication. The manuscript is extremely short with 3 out of 5 figures as supplemental data as well as part of the discussion as supplemental.

Although there is a fairly amount of studies showing that cardiac glycosides have anti-viral effects and there is a manuscript studying the role of cardiac glycosides in Dengue Virus, another Flavivirus (Cheung, Antiviral Res, 2014), the present manuscript shows intriguing in vivo data. However, the correlation between the in vitro and the in vivo data in regards to the action of cardiac glycosides is not clear, mechanistic data is lacking and for the results presented, both effects could be by different mechanisms.

1. The present manuscript does not recognize in the introduction or in the discussion the fact that cardiac glycosides have been already determined to have anti-viral effects in different virus families.

Response:

Thank you for your valuable suggestion. We added “the fact that cardiac glycoside have been already determined to have anti-viral effects in different virus families” in

the discussion of the manuscript, please kindly check it.

Moreover, in our study (data not shown), we demonstrated that both ouabain and digoxin had pan-antiviral effects on flaviviruses, including JEV, ZIKV, DENV, YFV and WNV. Additionally, we studied the antiviral effects on other types of viruses, such as the negative-sense RNA virus vesicular stomatitis virus (VSV), bi-segment negative-sense RNA virus lymphocytic choriomeningitis virus (LCMV), and positive-sense RNA virus human enterovirus 71 (EV71), both ouabain and digoxin had extended their antiviral effects to these viruses. The data are shown below.

2. The mechanism of action of the drugs is important to know. The introduction lacks some explanation about the virus cycle, so the reader can understand in what step the drug might act.

Response:

Thank you for your authoritative advice.

To date, there is no specific approved antiviral drug combating flaviviruses, thus necessitating the development of effective compounds and the exploration of novel antiviral targets. Flaviviruses contain an approximately 11-kb positive-stranded RNA genome that encodes three structural proteins and seven nonstructural (NS) proteins. The envelope glycoprotein (E) is the natural target for antibodies and entry inhibitors designed to prevent receptor-binding and membrane fusion (Yu Y. *Nature communications*. 2017; Chen L. *Antiviral Res.* 2017; Zhao H. *Cell*. 2016; Wang Q. *Science translational medicine*. 2016). Likewise, viral proteases such as NS2B-NS3 protease-helicase, and the NS5 RNA-dependent RNA polymerase, represent attractive drug targets for identifying replication inhibitors (Luo D. *Antiviral Res.* 2015; Sampath A. *Antiviral Res.* 2009). Besides these viral targets, cellular proteins involved in the life cycle also serve as attractive therapeutic targets. During infection, flaviviruses are internalized by receptor-mediated endocytosis, and complete transcription and translation, as well as assembly and budding at the endoplasmic reticulum (ER) membrane (Fernandez-Garcia MD. *Cell Host Microbe*. 2009). Accordingly, drugs that inhibit the acidification of endosomes and reduce ER stress are proven to be effective as anti-flavivirus agents both *in vitro* and *in vivo* (Gladwyn-Ng I. *Nature neuroscience*. 2018; Li C. *EBioMedicine*. 2017). Similarly, inhibitors targeting the host ubiquitin-protease system, autophagic activity, and interferon (IFN) signaling pathway were found to be effective in preventing flavivirus

infection (Wang P. *Virology*. 2016; Li C. *Immunity*. 2017; Cao B. *J Exp Med*. 2017).

Zika virus is an arbovirus that belongs to the *Flavivirus* genus and *Flaviviridae* family.

The genome of ZIKV is positive sense linear, with monopartite RNA of approximately 11 kb in length. The genome encodes for a polyprotein that is cleaved by host and viral proteases into three structural proteins (capsid, pre-membrane, and envelope) and seven non-structural proteins (NS1, NS2A, NS2B, NS3, NS4A, NS4B, and NS5) (Cecile Baronti. *Genome Announcements*. 2013). The structural proteins play roles in virus particle formation, receptor binding, virus fusion, and host cell entry; the non-structure proteins are responsible for viral genome replication and host immunity evasion (Grant A. *Cell Host Microbe*. 2016; Kumar A. *EMBO report*. 2016).

In humans, ZIKV infection typically causes a self-limiting and mild illness known as Zika fever, which often is accompanied by myalgia, maculopapular rash, and headache. Human dermal fibroblasts, epidermal keratinocytes and immature dendritic cells were found to be permissive to ZIKV infection. The AXL, DC-SIGN, Tyro, and TIM-1 entry/adhesion factors permit entry of ZIKV. ZIKV replication activates the production of type I interferon and an antiviral immune response in infected cells. The formation of autophagosomes is connected with enhanced viral replication, and the induced expression of antiviral antigen clusters (RIG-1, MDA-5, and TLR3) that are able to detect the presence of pathogen-associated molecular patterns was observed after infection of skin fibroblasts (Hamel R. *J Virol*. 2015).

We added some explanations about the ZIKV cycle in the introduction of the manuscript, please kindly check it.

3. *Cardiac glycosides are toxic (being the Na,K-ATPase a housekeeping protein); the authors should show time of incubation with ouabain and digoxin in supplemental figure 1 as well as cell death at that time.*

Response:

Following your advice, we show the time of incubation (48 h) with ouabain and digoxin in supplemental figure 1 (figure 1 in the revised manuscript), and we calculated the IC₅₀s (digoxin: IC₅₀=91.82 nM; ouabain: IC₅₀=48.39 nM) of two compounds by IFA assay. In the experiment, Vero cells in 96-well plates were infected with ZIKV strain H/PF/2013 at MOI 0.8 in the presence of different concentrations for 48 h. Infection was stopped by rinsing each well once with PBS and fixing the cells with 4% paraformaldehyde. Cells were permeabilized using PBS with 0.2% Triton X-100 for 20 min and blocked with 5% FBS (Gibco), followed by treatment with primary antibody anti-ZIKV NS3 overnight at 4°C. After six rinses with PBS, were stained with DyLight™ 488 labeled antibody to rabbit IgG (KPL, Gaithersburg, MD, USA). Nuclei were counterstained by DAPI (Sigma-Aldrich, USA). Nine fields per well were imaged on an Operetta high-content imaging system (PerkinElmer) and the percentages of infected and DAPI-positive cells were calculated using the associated Harmony 3.5 Software. The inhibitory effect was evaluated as: $\text{Inhibition}\% = \left(1 - \frac{(\text{infected cells}/\text{DAPI cells})_{\text{compound}}}{(\text{infected cells}/\text{DAPI cells})_{\text{DMSO}}}\right) \times 100\%$. Three independent experiments were performed in duplicate for the calculation of IC₅₀ using GraphPad Prism 8 software. Dose-response curves of

ouabain and digoxin for inhibition of ZIKV infection are shown as below.

Moreover, we performed MTT assay to evaluate the cytotoxicity of two compounds.

However, CC_{50} value cannot be successfully fitted from toxicity curves. We reviewed the literatures reporting the cytotoxicity caused by ouabain and listed the reported data in the table below. Notably, the CC_{50} value varies in a wide range on different cell types, and for the same cell type, this value decreases sharply as the time extended from 24 h to 72 h (8).

CC_{50} (μM)	cell type	method	literature
>100	murine embryonic stem cells (mESCs)	MTT	(1)
>300	porcine trabecular meshwork (TM) cells	LDH	(2)
>300	murine glial cells	MTT, LDH	(3)
10,000*	1321N1 (Glial-like cell line derived from a human brain astrocytoma)	MTT	(4)
>200	BHK-21	Growth experiments	(5)
>10**	Vero	WST-1	(6)
15.11 [#]	Vero	MTT	(7)
<0.01 ~ >1 ^{##}	adrenocortical tumor cells	MTT	(8)

*only 1 mM was tested in the literature; ** 10 μM was the maximum tested concentration; [#] CC_{50} value was tested after a 72 hour incubation time; ^{##} CC_{50} varies in a wide range with different cell types for different incubation time.

1. Lee YK, Ng KM, Lai WH, Man C, Lieu DK, Lau CP, Tse HF, Siu CW. 2011. Ouabain facilitates

- cardiac differentiation of mouse embryonic stem cells through ERK1/2 pathway. *Acta pharmacologica Sinica* **32**:52-61.
2. **Dismuke WM, Mbadugha CC, Faison D, Ellis DZ.** 2009. Ouabain-induced changes in aqueous humour outflow facility and trabecular meshwork cytoskeleton. *The British journal of ophthalmology* **93**:104-109.
 3. **Kinoshita PF, Yshii LM, Orellana AMM, Paixao AG, Vasconcelos AR, Lima LS, Kawamoto EM, Scavone C.** 2017. Alpha 2 Na⁽⁺⁾,K⁽⁺⁾-ATPase silencing induces loss of inflammatory response and ouabain protection in glial cells. *Scientific reports* **7**:4894.
 4. **Ghasemi S, Salarian AA, Zare Mirakabadi A, Jafarnejad S, Ghazi-Khansari M.** 2017. Effect of Crude Venom of *Odonthobuthus doriae* Scorpion in Cell Culture using Ion Channel Modulators. *Iranian journal of pharmaceutical research : IJPR* **16**:648-652.
 5. **McDonald TF, Sachs HG, Orr CW, Ebert JD.** 1972. Multiple effects of ouabain on BHK cells. *Experimental cell research* **74**:201-206.
 6. **Dodson AW, Taylor TJ, Knipe DM, Coen DM.** 2007. Inhibitors of the sodium potassium ATPase that impair herpes simplex virus replication identified via a chemical screening approach. *Virology* **366**:340-348.
 7. **Su CT, Hsu JT, Hsieh HP, Lin PH, Chen TC, Kao CL, Lee CN, Chang SY.** 2008. Anti-HSV activity of digitoxin and its possible mechanisms. *Antiviral Res* **79**:62-70.
 8. **Pezzani R, Rubin B, Redaelli M, Radu C, Barollo S, Cicala MV, Salva M, Mian C, Mucignat-Caretta C, Simioni P, Iacobone M, Mantero F.** 2014. The antiproliferative effects of ouabain and everolimus on adrenocortical tumor cells. *Endocrine journal* **61**:41-53.

In the reference 7 above, the cytotoxicity of digoxin and ouabain in Vero cells after 72 h of incubation were determined by MTT assay. The authors reported CC₅₀s value of two compounds: digoxin: CC₅₀= 10.21 μM; ouabain octahydrate: CC₅₀=15.11 μM.

As mentioned above, the CC₅₀ value of two compounds decrease sharply as the time extended from 24 h to 72 h for the same cell type. Through comprehensive analysis of literature and data from our experiment, we concluded that digoxin: CC₅₀>10.21 μM; ouabain : CC₅₀>15.11 μM after 48 h of incubation, and ZIKV inhibition by ouabain and digoxin are not attributable to cytotoxicity, indicating a substantial difference between therapeutic and toxic concentrations.

4. *What is the effect of ouabain in the mice? Do they get sick? Did the authors*

measure weight loss?

Response:

In our mouse experiment, ouabain had no obvious toxicity on the mice, the mice did not get sick and their body weight, hair and a series of physiological phenomena are normal. We measured body weight of three mice models to evaluate the therapeutic administration in our study, the data are as follows:

A. Ouabain administration to C57BL/6 background pregnant mice, the mice were intragastrically administered with ouabain at 3 mg/kg of body weight for 6 days.

B. Ouabain administration to BALB/c mice, the mice were intraperitoneally administered with ouabain at 3 mg/kg of body weight for 21 days.

And we also detected creatinine, AST and ALT in serum, three indicators are normal, the data are shown below.

C. Ouabain administration to *Ifnar*^{-/-} mice, the mice were intraperitoneally administered with ouabain at 2 mg/kg of body weight for 6 days.

5. It is known that these drugs inhibit protein translation at certain dose. Is this the mechanism that the authors are suggesting? Please, explain and demonstrate if that is the case or another mechanism is proposed.

Response:

Thank you for your informative comments.

As there are four isoforms of the α subunit ($\alpha 1-4$) and three isoforms of the β subunit ($\beta 1-3$) of Na^+/K^+ -ATPase, and the $\alpha 2$ serves as the important target of cardiac glycosides. We firstly review the binding affinity of ouabain and digoxin to the $\alpha 2$. It is reported that ouabain and digoxin bind to $\alpha 2$ with the similar K_D values of 29 nM and 25.6 nM, respectively. Ouabain and digoxin also bind to other isoforms with different binding affinity ranging within a nanomolar level (Katz. J Biol Chem. 2015).

In the current work, we performed “addition of sodium and potassium” assay (figure 3

in the revised manuscript) and concluded that ZIKV inhibition with digoxin and ouabain occurs via Na^+/K^+ - ATPase. We proposed another mechanism that may contribute to the antiviral effects, ouabain may block ZIKV infection by induce cellular stress response. Ouabain treatment in mammalian cells causes the interaction between the inositol 1,4,5- trisphosphate (IP3) receptor and Na^+/K^+ - ATPase to induce calcium oscillations, further activate calcium dependent transcription factors, such as the nuclear factor (NF)-kappaB and activator protein(Aizman o. Proceedings of the National Academy of Sciences of the United States of America. 2001). And some evidences suggested that cardiac glycosides mediate inflammatory processes via the activation of the phosphoinositide 3-kinase/Akt pathway and the Src/mitogen-activated protein kinase pathway (Liu. Frontiers in physiology. 2016; Orellana AM. Frontiers in endocrinology. 2016). These signaling pathways also cause in the activation of NF- kappaB. Many of gene expressed were elicited by activation of NF- kappaB, including innate immune response, growth and differentiation that stimulate an antiviral state that maybe block ZIKV infection in the end. (Silverman NS. Gene & Developmet. 2001).

6. Supplemental Figure 3 is very confusing. The cells were pre-treated with different amounts of Na^+ or K^+ previous infection. What is the rational? Zero extracellular K^+ should have the same result as Na,K -ATPase inhibition, as it is the substrate for the pump to work. If I understand the graph correctly at zero extracellular K^+ , there is no replication inhibition.

Response:

Firstly, we are sorry for the confusion as the Y axis in supplemental Figure 3 (Figure 3 in the revised manuscript) is the “percentage of viral titer relative to control”. We have revised the Y axis title.

Secondly, we carefully review your comments and the data in figure 3. The rationale of the cells were pre-treated with different amounts of Na^+ or K^+ previous infection is shown below: Based on our experiment, a 49 h incubation period (in the current work) with the additional of NaCl and KCl causes little effect on cell viability, while a longer time incubation (5-21 days) might lead to damages (Stubblefield. Cancer Res. 1960; Dmitriev. Cell Cycle. 2007). Moreover, the viral titers obtained from the supernatants with increasing concentration NaCl or KCl are similar to the titers obtained from the normal medium (figure below). In figure 3, the effects observed when changing the concentrations of Na^+ and K^+ ions are relatively low when used with some concentration of digoxin or ouabain, while the effects are robust when used with some concentration (such as 49 nM) of digoxin or ouabain. In general, the ZIKV inhibitory effects of both compounds are enhanced by addition of the extracellular NaCl and alleviated by KCl, which indicate that the inhibition occurs via antagonism of Na^+/K^+ - ATPase.

The effects of additional sodium and potassium on ZIKV infection. Vero cells seeded in 96-well plate were incubated with increasing concentration of NaCl and KCl for 1 h, respectively. Then ZIKV H/PF/2013 strain with an MOI of 0.8 were added in each well. After 1 h, the cell supernatant were removed and treated with the indicated concentration of NaCl or KCl for an additional 47 h, and the virus titer were determined by plaque assay using Vero cells.

Moreover, we have employed another inhibitor, obatoclax, which inhibited flavivirus by blocking the membrane fusion, as a control. As shown in the figure below, 0.1 μ M obatoclax could result in a ~45% inhibition in ZIKV infection, and the excess Na⁺ or K⁺ had little effect on the inhibition. These data further verify that antagonism of Na⁺/K⁺ ATPase is the mechanism by which ouabain and digoxin inhibits ZIKV.

The effects of additional sodium and potassium on obatoclox inhibition. Vero cells were treated with 0.1 μM obatoclox diluted in medium supplemented with increasing concentration of NaCl or KCl for 1h, respectively. Then ZIKV H/PF/2013 strain with an MOI of 0.8 were added in each well. After 1 h, the cell supernatant were removed and treated with 0.1 μM obatoclox in ion-supplemented medium for an additional 47 h, and the virus titer were determined by plaque assay using Vero cells. Inhibition rate are calculated as percentage of infected cells normalized to DMSO-treated cells for duplicate experiments.

Thirdly, at the same concentration of digoxin, ZIKV inhibition reduced with the decrease of NaCl (figure 3a) but the increasing of KCl (figure 3c). Under the same experimental conditions, the results of ouabain action were consistent with digoxin (figure 3b and 3d), which indicated that the ZIKV-inhibiting effect of both compounds is positively correlated with NaCl, but inversely correlated with KCl. At zero extracellular K⁺, digoxin and ouabain inhibited in a dose-responsive manner, Therefore, the ZIKV inhibitory effect of both compounds is enhanced by addition of

the extracellular NaCl and alleviated by KCl.

We hope that your concerns have been adequately addressed.

7. The authors claim that the effect observed in the brain is due to Na,K-ATPase alpha2 and alpha 3 inhibition. However, all the effects studied in vitro were in cells that express almost exclusively the alpha1 subunit. Could the mechanism be different with the results in vitro and in vivo?

Response:

Thank you for your informative comments.

To address it, we review the literatures reporting the effects of ouabain on CNS function and the mechanism be different with the results in vitro and in vivo. Ouabain is also known as a hormone produced endogenously by the adrenal gland and the hypothalamus (Schoner. *Exp Clin Endocrinol Diabetes*. 2000). In pregnant women, plasma concentration of endogenous ouabain is much higher than healthy individual, and keeping increased towards the delivery (Vakkuri. *J Endocrinol*. 2000). Recently, the increasing endogenous ouabain in pregnant animals was proved to have a protective effect on the fetal development and growth (Dvela-Levitt. *J Neurotrauma*. 2014, Dvela-Levitt. *J Neurotrauma*. *J Am Soc Nephrol*. 2015). On the other hand, the target of ouabain, Na⁺/K⁺-APTase are abundant in the brain. In the CNS, neurons express the ATPase α 1 and α 3 isoforms, while glial cells express the α 1 and α 2 isoforms. So far, due to lack of the direct evidence, it is still hard to conclude that ouabain could cross the normal intact BBB (Sugimoto. *Drug Metab Dispos*. 2011).

However, ZIKV is characterized by disruption of the BBB integrity and enhanced infiltration of immune cells into the CNS, which is supposed to confer the permeability of ouabain to cross BBB too.

8. *What Na,K-ATPase isoforms does the placenta express?*

Response:

Thank you again.

The Na⁺/K⁺-ATPase was first described 63 years ago by Jens Christian Skou (Skou, 1957). The Na⁺ and K⁺ gradients across the plasma membrane are used by animal cells for numerous processes, and the range of demands requires that the responsible Na⁺/K⁺-ATPase, ion pump, could be fine-tuned to the different cellular needs. Thus, several isoforms are expressed of each of three subunits (the alpha, beta and FXYD subunits) that make a Na⁺/K⁺ - ATPase. Plant cells have no endogenous Na⁺/K⁺-ATPase and they use a proton gradient to energize their membranes, so Na⁺/K⁺-ATPase inhibitors are not toxic to them (Michael V. Clausen. *Frontiers in Physiology*.2017).

In human tissues, we reviewed the literature and reported that Na⁺/K⁺-ATPase alpha 1, alpha 2 and alpha 3 express in the placenta. Moreover, Na⁺/K⁺ - ATPase alpha-isoform abundance in placental tissue from women not having labor and from women in active, spontaneous labor. The levels of alpha 1 isoform in placenta obtained from women in active labor compared with placenta obtained from nonlaboring subjects were not significantly different; For the placental alpha 2

isoform, level were slight lower in women who were in labor, but the difference was not statistically significant; However, the placental alpha 3 isoform levels of women in active labor were significantly lower than those of women who had not experienced labor (M. Sean Esplin. Am J Obstet Gynecol. 2003). Moreover, in recently report, very low levels of a transcript corresponding to the human and mouse alpha 4 orthologs also have been detected in heart, liver, pancreas and placenta (Sergei Keryanov, Gene, 2002).

Response to reviewers.

Review 2:

We sincerely appreciate your professional comments about our work. We have revised and improved our manuscript according to your advices. Our responses to the specific comments are as follows.

In this study, the authors found that inhibition of Na⁺/K⁺ ATPase by using ouabain (or digoxin) can block Zika virus (ZIKV) infection in both cell culture and mouse models. Although several studies have reported that those two inhibitors can impairs infections with different viruses such as Chikungunya, Japanese encephalitis virus and human cytomegalovirus with similar mechanisms, this manuscript is still may afford a promising therapeutic potential in combating ZIKV. The manuscript needs a minor revision before accepted.

1. Please calculate the EC50 value of each inhibitor and clarify the fact that ZIKV inhibition by ouabain (or digoxin) are not attributable to their cytotoxicity.

Response:

Thank you for your authoritative advice, we calculated the IC₅₀s (digoxin: IC₅₀=91.82 nM; ouabain: IC₅₀=48.39 nM) of two compounds by IFA assay and added the data in the figure 1 (figure 1 in the revised manuscript). In the experiment, Vero cells in 96-well plates were infected with ZIKV strain H/PF/2013 at MOI 0.8 in the presence

of different concentrations for 48 h. Infection was stopped by rinsing each well once with PBS and fixing the cells with 4% paraformaldehyde. Cells were permeabilized using PBS with 0.2% Triton X-100 for 20 min and blocked with 5% FBS (Gibco), followed by treatment with primary antibody anti-ZIKV NS3 overnight at 4°C. After six rinses with PBS, were stained with DyLight™ 488 labeled antibody to rabbit IgG (KPL, Gaithersburg, MD, USA). Nuclei were counterstained by DAPI (Sigma-Aldrich, USA). Nine fields per well were imaged on an Operetta high-content imaging system (PerkinElmer) and the percentages of infected and DAPI-positive cells were calculated using the associated Harmony 3.5 Software. The inhibitory effect was evaluated as: $\text{Inhibition\%} = (1 - (\text{infected cells/DAPI cells})_{\text{compound}} / (\text{infected cells/DAPI cells})_{\text{DMSO}}) \times 100\%$. Three independent experiments were performed in duplicate for the calculation of IC₅₀ using GraphPad Prism 8 software. Dose-response curves of ouabain and digoxin for inhibition of ZIKV infection are shown below.

Moreover, we performed MTT assay to evaluate the cytotoxicity of two compounds. However, CC₅₀ value cannot be successfully fitted from toxicity curves. We reviewed the literatures reporting the cytotoxicity caused by ouabain and list the reported data in the table below. Notably, the CC₅₀ value varies in a wide range on different cell

types, and for the same cell type, this value decreases sharply as the time extended from 24 h to 72 h (8).

CC ₅₀ (μM)	cell type	method	literature
>100	murine embryonic stem cells (mESCs)	MTT	(1)
>300	porcine trabecular meshwork (TM) cells	LDH	(2)
>300	murine glial cells	MTT, LDH	(3)
10,000*	1321N1 (Glial-like cell line derived from a human brain astrocytoma)	MTT	(4)
>200	BHK-21	Growth experiments	(5)
>10**	Vero	WST-1	(6)
15.11 [#]	Vero	MTT	(7)
<0.01 ~ >1 ^{###}	adrenocortical tumor cells	MTT	(8)

* only 1 mM was tested in the literature; ** 10 μM was the maximum tested concentration; [#]CC₅₀ value was tested after a 72 hour incubation time; ^{###}CC₅₀ varies in a wide range with different cell types for different incubation time.

1. **Lee YK, Ng KM, Lai WH, Man C, Lieu DK, Lau CP, Tse HF, Siu CW.** 2011. Ouabain facilitates cardiac differentiation of mouse embryonic stem cells through ERK1/2 pathway. *Acta pharmacologica Sinica* **32**:52-61.
2. **Dismuke WM, Mbadugha CC, Faison D, Ellis DZ.** 2009. Ouabain-induced changes in aqueous humour outflow facility and trabecular meshwork cytoskeleton. *The British journal of ophthalmology* **93**:104-109.
3. **Kinoshita PF, Yshii LM, Orellana AMM, Paixao AG, Vasconcelos AR, Lima LS, Kawamoto EM, Scavone C.** 2017. Alpha 2 Na⁽⁺⁾,K⁽⁺⁾-ATPase silencing induces loss of inflammatory response and ouabain protection in glial cells. *Scientific reports* **7**:4894.
4. **Ghasemi S, Salarian AA, Zare Mirakabadi A, Jafarnejad S, Ghazi-Khansari M.** 2017. Effect of Crude Venom of *Odonthobuthus doriae* Scorpion in Cell Culture using Ion Channel Modulators. *Iranian journal of pharmaceutical research : IJPR* **16**:648-652.
5. **McDonald TF, Sachs HG, Orr CW, Ebert JD.** 1972. Multiple effects of ouabain on BHK cells. *Experimental cell research* **74**:201-206.
6. **Dodson AW, Taylor TJ, Knipe DM, Coen DM.** 2007. Inhibitors of the sodium potassium ATPase that impair herpes simplex virus replication identified via a chemical screening approach. *Virology* **366**:340-348.
7. **Su CT, Hsu JT, Hsieh HP, Lin PH, Chen TC, Kao CL, Lee CN, Chang SY.** 2008. Anti-HSV activity of digitoxin and its possible mechanisms. *Antiviral Res* **79**:62-70.

8. Pezzani R, Rubin B, Redaelli M, Radu C, Barollo S, Cicala MV, Salva M, Mian C, Mucignat-Caretta C, Simioni P, Iacobone M, Mantero F. 2014. The antiproliferative effects of ouabain and everolimus on adrenocortical tumor cells. *Endocrine journal* **61**:41-53.

In the reference 7 above, the cytotoxicity of digoxin and ouabain in Vero cells after 72 h of incubation were determined by MTT assay. The authors reported CC_{50} s value of two compounds: digoxin: $CC_{50}= 10.21 \mu\text{M}$; ouabain octahydrate: $CC_{50}=15.11 \mu\text{M}$.

As mentioned above, the CC_{50} value of two compounds decrease sharply as the time extended from 24 h to 72 h for the same cell type. Through comprehensive analysis of literature and data from our experiment, we concluded that digoxin: $CC_{50}>10.21 \mu\text{M}$; ouabain : $CC_{50}>15.11 \mu\text{M}$ after 48 h of incubation, and ZIKV inhibition by ouabain and digoxin are not attributable to cytotoxicity, indicating a substantial difference between therapeutic and toxic concentrations.

2. It was reported that cardiac glycosides like ouabain and digoxin could inhibit Na^+/K^+ ATPase by binding to the catalytic subunit but did so with less efficiency to specific murine isoforms relative to their human counterparts. Considering the author used the mouse model in this study, please use some murine cell lines to clarify the possible species-specific inhibition effect of ouabain and digoxin providing useful information for justifying the dose usage (2 mg/kg) of ouabain in mouse model.

Response:

Thank you for your informative comments.

In our current work, we performed “addition of sodium and potassium” assay (figure 3 in the revised manuscript) and concluded that ZIKV inhibition with digoxin and

ouabain occurs via Na^+/K^+ - ATPase.

As your valuable comments, ouabain and digoxin could inhibit Na^+/K^+ - ATPase by binding to the catalytic subunit but did so with less efficiency to specific murine isoforms relative to their human counterparts. We also concerned about the antiviral effect of ouabain on mice before doing animal experiment, and we used BHK-21 cells to clarify the possible species-specific inhibition effect of ouabain. BHK -21 cells electroporated with the flavivirus replicons were treated with 10 μM ouabain respectively, we revealed that appreciable reduction of luciferase signal at 24 h postelectroporiton to evaluate the antiviral effect of ouabain, the result are shown below indicated that 10 μM ouabain has no effect on flavivirus infected BHK-21 cells. Moreover, the literatures reported that CC_{50} value of ouabain $> 200 \mu\text{M}$ on BHK-21 cells (McDonald. Experiment cell research. 1972), the insensitivity of cells to ouabain indirectly affected the antiviral effect.

However, many ZIKV pathogenesis studies current are being conducted in mice, and

rodents are considered first-line animal models for antiviral drug testing, demonstration of efficacy in small animal models is generally a prerequisite for therapeutic testing in non-human primates and human volunteers. On the other hand, ouabain shows strong inhibitory activity on ZIKV in vitro experiment and identifies the Na^+/K^+ ATPase as a promising antiviral target for treatment of ZIKV infections. Therefore, we tried to find a proper concentration which is not only antiviral but also no toxicity on two ZIKV infected mice models (models A and B) and a common animal model (model C), the data are shown below:

A. Ouabain administration to C57BL/6 background pregnant mice, the mice were intragastrically administered with ouabain at 3 mg/kg of body weight for 6 days.

B. Ouabain administration to *Ifnar^{-/-}* mice, the mice were intraperitoneally administered with ouabain at 2 mg/kg of body weight for 6 days.

C. Ouabain administration to BALB/c mice, the mice were intraperitoneally administered with ouabain at 3 mg/kg of body weight for 21 days.

And we also detected creatinine, AST and ALT in serum, three indicators are normal, the data are shown below.

The weight curves show that the administrations of 3 mg/kg (i.g. model A) and 2mg/kg (i.p. model B) ouabain had no toxicity on mice, and significantly reduced ZIKV-associated injuries in the fetal brain, thus prevent the development of microcephaly. Meanwhile, ouabain also reduced the viral load in the serum and brain of *Ifnar^{-/-}* mice. Based on this result, we review the literatures reporting the effects of ouabain on CNS function. Ouabain is also known as a hormone produced endogenously by the adrenal gland and the hypothalamus (Schoner. Exp Clin

Endocrinol Diabetes. 2000). In pregnant women, plasma concentration of endogenous ouabain is much higher than healthy individual, and keeping increased towards the delivery (Vakkuri. J Endocrinol. 2000). Recently, the increasing endogenous ouabain in pregnant animals was proved to have a protective effect on the fetal development and growth (Dvela-Levitt. J Neurotrauma. 2014, Dvela-Levitt. J Neurotrauma. J Am Soc Nephrol. 2015). On the other hand, the target of ouabain, Na⁺/K⁺-ATPase are abundant in the brain. In the CNS, neurons express the ATPase α 1 and α 3 isoforms, while glial cells express the α 1 and α 2 isoforms. So far, due to lack of the direct evidence, it is still hard to conclude that ouabain could cross the normal intact BBB (Sugimoto. Drug Metab Dispos. 2011). However, ZIKV is characterized by disruption of the BBB integrity and enhanced infiltration of immune cells into the CNS, which is supposed to confer the permeability of ouabain to cross BBB too.

3. The author proposed ouabain inhibit ZIKV infection at the replication stage, but they only checked the expression of the viral structural protein NS3. Please determine the fate of some other protein machineries involved in different steps of replication stage like NS1 before and after administration of inhibitors or even generating or using some ouabain (or digoxin)-resistant viral mutants to give more clues about the detailed mechanism of those inhibitors.

Response:

We do appreciate your professional comments and checked the expression of the ZIKV viral structural protein NS5 (we are sorry our lab don't have NS1 antibody).

The IFA data are shown below:

In our study, we demonstrated that both ouabain and digoxin had pan-antiviral effects on flaviviruses, including JEV, ZIKV, DENV, YFV, and WNV. Next, we studied the antiviral effects on other types of viruses, such as the negative-sense RNA virus vesicular stomatitis virus (VSV), bi-segment negative-sense RNA virus lymphocytic choriomeningitis virus (LCMV), and positive-sense RNA virus human enterovirus 71 (EV71), the data are shown below. The relatively broad-spectrum antiviral activity of ouabain and digoxin suggests that both drugs act via targeting cellular factors rather than viral proteins. Notably, we tried to select drug-resistant variants by serial passaging of JEV and ZIKV using increasing concentrations of digoxin or ouabain, respectively. However, no adaptive mutant virus was found after 25 passages of either virus. This result is consistent with the hypothesis that these drugs act by targeting cellular protein, making the barrier to resistance more difficult to overcome.

Thank you again for your valuable advice.

Reviewers' comments:

Reviewer #1 (Remarks to the Author):

I am afraid the authors did not answer my questions appropriately:

Cytotoxicity was not evaluated. The authors reply with a literature review. It is important to show that the effects the authors see at 48h are unrelated to cell death (Figure 2) as well that as the higher does in Figure 1. The IC50 in Figure 1k does not correspond with the images in Figure 1i. Are cells death when 392 nM is used?

No hypothesis of mechanism is proposed in the new version of the experiment. The authors show that there is inhibition of RNA synthesis. The rebuttal letter suggests that the mechanism is cellular stress response. Not only their explanation does not make sense explaining the decreased RNA synthesis, but experiments should be done to suggests this theory. The doses of the IC50 for these drugs (Figure 1) would suggest that that is not the case as usually these drugs works with the Na,K-ATPase as receptor in lower doses.

Reviewer #2 (Remarks to the Author):

Although the authors can not address the detailed mechanism of the antiviral effects of both ouabain and digoxin, I still recommend publishing this manuscript since their animal model is impressive and may trigger further in vivo studies about those two inhibitors.

Response to Reviewers

Dear Reviewer 1:

Thank you for the time and effort that have put into reviewing the previous version of the manuscript, your professional suggestion have enabled us to improve our work.

Our responses to the specific comments are as follows.

COMMENTS TO THE AUTHOR:

1, Cytotoxicity was not evaluated. The authors reply with a literature review. It is important to show that the effects the authors see at 48h are unrelated to cell death (Figure 2) as well that as the higher does in Figure 1. The IC50 in Figure 1k does not correspond with the images in Figure 1i. Are cells death when 392 nM is used?

Response:

Thank you for your authoritative advice.

We performed CCK8 assay, MTT assay and DAPI assay to evaluate the cytotoxicity of two drugs.

CCK8 and MTT assay were carried out according to the instructions. In the DAPI assay, Vero cells in 96-well plates were incubated with two compounds of different concentrations for 48 h. Incubation was stopped by rinsing each well once with PBS and fixing the cells with 4% paraformaldehyde. Cells were permeabilized using PBS with 0.2% Triton X-100 for 20 min and followed by treatment with three rinses with PBS, nuclei were stained by DAPI (Sigma-Aldrich, USA). Nine fields per well were imaged on an Operetta high-content imaging system (PerkinElmer) and the

percentages of DAPI-positive cells were calculated using the associated Harmony 3.5 Software. The cell viability was evaluated as: $\text{Cell viability \%} = \frac{(\text{DAPI cells})_{\text{compound}}}{(\text{DAPI cells})_{\text{DMSO}}} \times 100\%$, the data are as follows:

Due to difference in detection methods, the percents of cell viability were slightly different. After taking all factors into consideration, we concluded that digoxin: $CC_{50} > 100 \mu\text{M}$; ouabain : $CC_{50} > 100 \mu\text{M}$ after 48 h of incubation in Vero cells. Moreover, The IC_{50} in Figure 1k was also calculated using the associated Harmony 3.5 Software under the standard condition, the data are consistent with the trend reflected in the IFA images, the ZIKV inhibition effects of ouabain and digoxin are not attributable to cytotoxicity in our experiment, indicating a large difference between their therapeutic and toxic concentrations.

COMMENTS TO THE AUTHOR:

2、 No hypothesis of mechanism is proposed in the new version of the experiment. The authors show that there is inhibition of RNA synthesis. The rebuttal letter suggests that the mechanism is cellular stress response. Not only their explanation does not make sense explaining the decreased RNA synthesis, but experiments should be done to suggest this theory. The doses of the IC_{50} for these drugs (Figure 1) would suggest that that is not the case as usually these drugs works with the Na,K-ATPase as receptor in lower doses.

Response:

Thank you again.

In our study, time-of-addition experiment was performed to investigate whether two drugs blocked the entry step or the replication step, we found that no suppression of viral titer by two drugs was observed when they were used as treatments before

infection or as a virucide, suggesting that two drugs do not inhibit ZIKV infection either by inactivating the virus directly or by blocking ZIKV entry. However, two drugs exerted fully inhibitory effects when they were added at 1 h postinfection, suggesting that viral replication was the stage at which two drugs showed inhibitory activity, and the inhibitory effects of ouabain and digoxin on ZIKV replicons confirmed it.

As there are four isoforms of the α subunit ($\alpha 1-4$) and three isoforms of the β subunit ($\beta 1-3$) of Na^+/K^+ -ATPase, and the $\alpha 2$ serves as the important target of cardiac glycosides. We firstly reviewed the binding affinity of ouabain and digoxin to the $\alpha 2$. It is reported that ouabain and digoxin bind to $\alpha 2$ with the similar K_D values of 29 nM and 25.6 nM, respectively (Katz. J Biol Chem. 2015), and our experimental concentrations ranged from nanomoles to micromoles. Some RNA viruses (Chikungunya virus, Coronavirus and lymphocytic choriomeningitis virus) have been reported to be inhibited by similar concentrations of ouabain and digoxin, the order of magnitudes is consistent with our experimental results.

We hope that your concerns have been adequately addressed.

Dear Reviewer 2:

Thank you for your encouragement of our work, we are so grateful for your valuable advice from the bottom of our heart.

Thanks very much for taking your time to review this manuscript !

REVIEWERS' COMMENTS:

Reviewer #1 (Remarks to the Author):

The authors answered all my concerns